# Surveillance of cfDNA Hot Spot Mutations in NSCLC Patients during Disease Progression

**DOI:** 10.3390/ijms24086958

**Published:** 2023-04-09

**Authors:** Agne Sestokaite, Vaida Gedvilaite, Saulius Cicenas, Rasa Sabaliauskaite, Sonata Jarmalaite

**Affiliations:** 1National Cancer Institute, Santariskiu 1, 08406 Vilnius, Lithuania; 2Institute of Biosciences, Life Sciences Center, Vilnius University, Sauletekio Ave. 7, 10257 Vilnius, Lithuania

**Keywords:** non-small cell lung cancer (NSCLC), liquid biopsy, *TP53*, *KRAS*, cell-free DNA (cfDNA) mutation load

## Abstract

Non-small cell cancer (NSCLC) has been identified with a great variation of mutations that can be surveyed during disease progression. The aim of the study was to identify and monitor lung cancer-specific mutations incidence in cell-free DNA as well as overall plasma cell-free DNA load by means of targeted next-generation sequencing. Sequencing libraries were prepared from cell-free DNA (cfDNA) isolated from 72 plasma samples of 41 patients using the Oncomine Lung cfDNA panel covering hot spot regions of 11 genes. Sequencing was performed with the Ion Torrent™ Ion S5™ system. Four genes were detected with highest mutation incidence: *KRAS* (43.9% of all cases), followed by *ALK* (36.6%), *TP53* (31.7%), and *PIK3CA* (29.3%). Seven patients had co-occurring *KRAS* + *TP53* (6/41, 14.6%) or *KRAS* + *PIK3CA* (7/41, 17.1%) mutations. Moreover, the mutational status of *TP53* as well an overall cell-free DNA load were confirmed to be predictors of poor progression-free survival (HR = 2.5 [0.8–7.7]; *p* = 0.029 and HR = 2.3 [0.9–5.5]; *p* = 0.029, respectively) in NSCLC patients. In addition, *TP53* mutation status significantly predicts shorter overall survival (HR = 3.4 [1.2–9.7]; *p* < 0.001). We demonstrated that *TP53* mutation incidence as well as a cell-free DNA load can be used as biomarkers for NSCLC monitoring and can help to detect the disease progression prior to radiological confirmation of the status.

## 1. Introduction

Lung cancer is the second most frequent cancer and the leading cause of cancer death in 2020, representing approximately 1 in 10 (11.4%) cancers diagnosed and 1 in 5 (18.0%) deaths [1]. In Lithuania, with a population of 2.7 million, a rough number of 1500 new cases of lung cancer are diagnosed annually [2]. Non-small cell cancer (NSCLC) accounts for approximately 85% of all cases of lung cancer [3]. NSCLC is proven to have a great diversity of molecular variations and molecular profiling of advanced tumors, which has allowed for the identification of oncogenic drivers [4]. Moreover, new actionable mutations in lung cancer are being discovered that increase the clinical significance of genetic testing for treatment selection and outcome monitoring in NSCLC.

Most studies which identified mutation hot spots in lung cancer have been performed primarily to identify mutations that interfere with protein coding [5]. At least 70% of the patients with advanced lung cancer have actionable mutations. The list includes *KRAS*, *EGFR*, and *ALK* as some of the most common, while *MET*, *BRAF*, *PIK3CA*, and others are more sparsely detected [6]. However, recurrent mutations in tumor suppressors, such as *TP53*, are also common, although the mutation status is more difficult to comprehend [7].

Nevertheless, diagnostic tumor sample collection is an invasive procedure and often inadmissible for patients due to the location of the tumor or health conditions [8]. For this group of patients, liquid biopsy can be applied. Liquid biopsy is minimally invasive, easily accessible, and can be routinely used for mutation detection in cancer diagnostics and research [9]. It is especially useful for disease monitoring as cell-free DNA (cfDNA) is released from apoptotic or necrotic tumor tissue cells. Using current molecular detection techniques, cfDNA fragments can be recognized as less than 100 bp long, released in blood, saliva, pleural effusion, and other biological fluids [10]. The “shedded” amount of cfDNA in bodily liquids depends on tumor cell mitotic rate, tumor volume, and vascular growth around the tumor [11]. As a result of these traits, cfDNA is a “real time” biomarker reflecting disease progression and growing tumor burden.

Recently, the number of studies with the purpose to evaluate blood-based mutational burden in NSCLC patients markedly increased, as it was shown that even rare genetic variants could lead to primary and secondary treatment resistance [12,13]. Furthermore, together with known oncogenic drivers, co-occurring mutations can significantly modify cellular processes and clinical outcomes [14,15,16]. A recent study with over 1100 patients with NSCLC demonstrated that, overall, the presence of cfDNA has the potential to be used as an independent overall survival predictor, with hazard ratios (HRs) reaching 2.05 and 95% confidence intervals (CIs) 1.74–2.42 (*p* < 0.001) [17]. Furthermore, increasing interest has been shown in the cfDNA dynamics during disease course, with a higher cfDNA amount detected after surgery for locally advanced NSCLC indicating shorter progression-free survival [18] or disease recurrence [19,20]. Overall, long-term tracking of tumor mutational burden could be a straightforward way to detect changes occurring during tumor progression and could assist in timely treatment modification. 

The aims of this study were to evaluate the plasma cfDNA mutation profile of most common lung cancer mutations in a representative Lithuanian NSCLC patient cohort and to analyze the prognostic value of targeted next-generation sequencing on liquid biopsy for outcome prediction.

## 2. Results

### 2.1. Circulating DNA Somatic Mutations

Somatic mutations were analyzed at 3 different timelines: T0 (before treatment), T1 (after treatment), and TX (during progression) in 11 gene hot spot regions and the regions surrounding these hot spots included in the Oncomine Lung cfDNA Assay gene panel: *ALK*, *BRAF*, *EGFR*, *ERBB2*, *KRAS*, *MAP2K1*, *MET*, *NRAS*, *PIK3CA*, *ROS1*, and *TP53*. In total, 40 different mutations were found in 8 different genes (*ALK*, *BRAF*, *EGFR*, *KRAS*, *MAPK2K1*, *MET*, *PIK3CA*, *TP53*) in all patient samples (*n* = 72). Out of all mutations, 33 were pathogenic, with the majority being SNVs (26/33, 78.8%), 5 indels (5/33, 15.2%), and 2 intronic/splice site variants (2/33, 6.1%). Pathogenicity was analyzed in silico and most of the variants were predicted to be deleterious (31/33, 93.9%), while only two variants were predicted to be benign. Out of all the pathogenic variants, after filtering for known pathogenic mutations, 26 variants (26/33, 78.8%) were of known pathogenicity described in the National Center for Biotechnology Information (NCBI) dbSNP or Catalogue of Somatic Mutations in Cancer (COSMIC) databases. One mutation described in the NCBI dbSNP database as affecting drug response (*EGFR* p.L858R) was also included in this group. Mutations outside the exon boundary were mostly located in the *ALK* gene, while the biggest mutation variation in locus was detected for the *TP53* gene. In addition, out of all detected benign mutations, 4 were described in the NCBI dbSNP or COSMIC databases (4/7, 57.1%). Most of the benign variants were SNVs (6/7, 85.7%) and 1 intronic variant (1/7, 14.3%). Only pathogenic variants were included in further analyses (Appendix A).

In the T0 group (*n* = 41), variants in cfDNA were not detected in 3 out of 41 samples, and the most frequently mutated gene was *KRAS* (43.9% of all cases), followed by *ALK* (36.6%), *TP53* (31.7%), *PIK3CA* (29.3%), *MAP2K1* (17.1%), *EGFR* (14.6%), *MET* (9.8%), and *BRAF* (2.4%). Median mutation count was 2 (95% CI: 1.0–3.0) per case. Moreover, concomitant mutations occurred in the most frequently mutated genes—*KRAS*, *TP53* and *PIK3CA*. More specifically, *KRAS* and *TP53* variants were found co-occurring in 6 blood samples (6/41, 14.6%) with 50.0% being adenocarcinomas, while *KRAS* and *PIK3CA* in 7 samples (7/41, 17.1%) co-occurred, of which 57.1% were squamous cell carcinomas (Figure 1).

### 2.2. Variation in cfDNA Mutation Count and VAF during the Disease Progression

In all serial samples (T1 and TX; *n* = 13 and *n* = 18) collected from 28 patients, at least one mutation was detected in 74.2% (23/31) plasma cfDNA samples, while 77.4% (24/31) of all serial samples had a mutation count of 3 or more. Moreover, 45.2% (14/31) had identical mutations in concordance to the T0 group (*n* = 41). Specifically, in the serial samples, the most common identical variants were detected in *KRAS*, *TP53*, *MET*, and *EGFR* genes (Appendix A).

In the T1 group, under half of the cases were with detected pathogenic mutation and had an average mutation count increase (from 1 to 3) as compared to paired samples from the T0 group. More specifically, 4 out of 13 (30.8%) cases were identified with the increased mutation count as compared to T0 (Figure 2A; patient Nos. 18, 28, 33 and 39). Of interest, patient No. 18 after treatment had 4 discordant mutations not detected in the paired T0 group samples: variant outside exon boundary in *ALK*; in *KRAS* gene, 2 SNVs (G15S and V7A); and a substitution in *MET* T1010I (Appendix A). Five out of 13 patients (38.5%) in the same group were identified with a mutation count decrease (from 3 to 1 mutations) after treatment (Figure 2A; patient Nos. 1, 2, 9, 26, and 38). In 13 patients with T1 serial samples, the median OS was longer (33.5 vs. 26.5 months) for cases with a mutation count decrease after treatment as compared to patients with a mutation count increase.

In the TX group, 33.3% had a mutation count increase from 2 to 4 (5/15, patient Nos. 4, 13, 15, 23, and 32) (Figure 2A). The highest variation of mutations in cfDNA not found in T0 (patient Nos. 13, 23, and 32) was in the *TP53* gene (Y205fs, M237I, V216M, and R158H) (Appendix A). No correlation with clinical–pathological characteristics was identified between patients with changes in mutation count in the T0 vs. TX samples.

The variant allele frequencies (VAFs) were compared in the serial samples of cfDNAs (*n* = 31) for the most common mutations. Overall, median VAFs were below 5% for the *KRAS*, *TP53*, *PIK3CA,* and *ALK* genes. The highest median VAF variation was identified for *KRAS* in the T0 vs. TX paired samples (4.9 vs. 6.0%; Figure 2B). A VAF higher than 5% was detected only in the TX group. For example, the *EGFR* L858R variant and non-frameshift insertion (patient Nos. 5 and 2), VAF increased more than 3 times during the disease progression (17.2 to 51.2 and 3.8 to 32.2%). For the *KRAS* gene, VAFs were consistently above 5% in all serial samples except for 3 patients (patient Nos. 20, 29, and 36) cfDNA, where variants in G12 position were 2 times higher (18.2 to 33.6; 4.8 to 20.8%) and variant Q61H was significantly depleted (22.6 to 0.2%) during progression. One patient (patient No. 15) had increasingly more elevated *TP53* C242Y variant during the disease progression (6.9 to 22.9%). However, an actionable variant V600E of *BRAF* in the serial samples was below 2% (patient No. 32), as well as two *TP53* missense variants and one nonsense variant (H178P, R175C, R213*) in 3 patients (patient Nos. 4, 24, 34) (Appendix A).

### 2.3. Variation in cfDNA Mutation Count and VAF during the Disease Progression

During the follow-up period (median 17.9 months, 95% CI: 13.6–20.7), the disease progression was diagnosed for 56.1% of the patients (23/41), and 65.9% (27/41) of the patients deceased. The impact of mutation status on overall and progression-free survival of NSCLC cases was evaluated by Cox proportional hazards models (Figure 3). The mutation status of either *TP53* R181P, H178P, C242Y, R175C, R280T, R249S, S241C, R213*, or G154fs variants was significantly associated with PFS and OS (HR = 2.6 [1.0–6.4]; *p* = 0.043 and HR = 3.7 [1.6–8.4]; *p* = 0.002, respectively). Patients with a mutation status of the *BRAF* V600E variant had a high tendency for disease progression (HR = 2.38 [0.31–18.01]; *p* = 0.404), while occurrence of mutated *EGFR*, *PIK3CA*, *ALK*, or *KRAS* did not significantly influence PFS or OS. 

Univariate and multivariate Cox regression models were performed. In the univariate analysis, only *TP53* mutation status was significantly associated with patient survival (Figure 3A,B).

According to the mutation count, NSCLC cases were distributed into the groups with 0 (*n* = 8) or 1–2 (*n* = 19) or ≥3 (*n* = 14) mutations. Mutation count did not impact PFS or OS significantly, although PFS and OS were several months shorter in the 1–2 mutation groups as compared to the 0 or ≥3 mutations’ groups (7.0 vs. 9.5 or 9.5 and 10.0 vs. 13.0 or 19.5 months, respectively) (Figure 4A,B). To be more precise, patients with the *TP53* mutation had significantly shorter median PFS and OS of 5.0 and 8.0 months compared to 14.0 and 24.5 months without the *TP53* mutation (HR = 2.5 [0.8–7.7]; *p* = 0.029 and HR = 3.4 [1.2–9.7]; *p* < 0.001) (Figure 4C,D). Conversely, in the multivariate model pTNM staging (III–IV vs. Ib-II, *p* = 0.032), the gene mutational status of *KRAS* (*KRAS* vs. N/A, *p* = 0.034), *TP53* (*TP53* vs. N/A, *p* = 0.001) and the mutational status of concomitant *KRAS* + *TP53* (*KRAS* + *TP53* vs. N/A, *p* = 0.001) were significantly associated with PFS. However, only the mutational status of *TP53* in the multivariate model was a significant predictor of OS (*TP53* vs. N/A, *p* = 0.003) (Appendix A).

Finally, we also weighed the impact of cfDNA load on PFS and OS. Specifically, NSCLC patients with a high cfDNA load (>50 mutant cfDNA molecules; *n* = 16) vs. a low cfDNA load (≤50 mutant cfDNA molecules, *n* = 25) had a higher risk of progression (HR = 2.3 [0.9–5.5]; *p* = 0.029), showing increasing tumor burden during the disease progression (Figure 5A). However, no influence on OS was observed between the high and low cfDNA load patient groups (HR = 1.1 [0.5–2.4]; *p* = 0.778) (Figure 5B).

## 3. Discussion

cfDNA plasma mutational signatures have been shown to be similar to the molecular profile of tumor foci [21,22]; thus, the cfDNA examination is especially useful when a tissue biopsy is difficult to obtain. Next-generation sequencing performed on blood cfDNA is recommended for advanced disease cases, and especially for routine *EGFR* and other actionable mutation testing [23]. Mutation status, mutation count, and mutation load in cfDNA have previously been shown as useful indicators for cancer progression [24,25] and could reflect systemic tumor burden and heterogeneity [26]. In the present study, we demonstrated that the most common pathogenic variants detectable in cfDNA from NSCLC patients are *KRAS*, *TP53*, and *PIK3CA* gene mutations. Further, the mutation status of *TP53* as well as the overall cfDNA load before treatment significantly predicted the PFS of NSCLC patients, while the mutation count did not. We also singled out the *TP53* mutation status as a predictor for shorter OS—as with any *TP53* mutation, the patients’ survival time was 3 times shorter than those without.

Frequently found in NSCLC, the gene *TP53* mutations are associated with the resistance to chemotherapy and shorter overall survival, although, due to the abundance of different mutations, it is difficult to find prospective targets for therapy. However, it is more common to look for concomitant actionable alterations that could be influenced by *TP53* or other driver mutations [27,28]. Often found together in non-smoking NSCLC patients, *EGFR* and *TP53* alterations have been identified as a prognostic implication of decreased survival [29,30]. Moreover, the interval of positive response to first-line *EGFR* tyrosine kinase inhibitors (TKIs) was also found to be reduced in patients with *TP53* or *KRAS* mutations in NSCLC [31,32]. In advanced disease, *TP53* and *KRAS* alterations are 50% more common in smokers vs. non-smokers [33]. *KRAS* mutant lung adenocarcinomas are often diagnosed with co-occurring *TP53* alterations, including loss or gain of function, that are usually associated with shorter PFS and increased metastasis probability [34,35]. We found that 14.6% of NSCLC patients were identified with concomitant *TP53* and *KRAS* mutations, of which 50.0% had histologically confirmed adenocarcinomas. A similar frequency of at least 12.8% of co-occurring *TP53* and *KRAS* mutations have been identified and were more common in heavy smokers with non-squamous lung cancer [36,37].

Since *KRAS* alterations are viewed as an old target, the role of *KRAS* in therapy should be reconsidered [38]—even more so, as combination therapy has been shown to increase overall and progression-free survival for patients when chemotherapy is not possible [39]. There is an increasing number of studies that consider the possibility of restoring normal protein function in mutant genes such as *KRAS* to account for the damaging effects of such mutations [40,41,42], particularly as there is increasing proof that co-occurring mutations with *KRAS* have significant clinical relevance. It has been shown that different *KRAS* mutation subtypes correlate with different concomitant mutations, such as coexistent *KRAS* (G12D) substitution with *PIK3CA* (H1047R), which could promote early *KRAS*-initiated tumorigenesis [43]. Moreover, it has been suggested that *KRAS* mutant NSCLC should not be overlooked as the most common oncogenic mutation in lung cancer, as incidences of such mutations have also been identified in subclonal populations, therefore indicating that the *KRAS* mutant NSCLC should be analyzed as a completely different subgroup of lung cancers [44]. In the present study, coexisting *KRAS* and *PIK3CA* mutations were found in 17.1% of patients; in more than half with squamous cell carcinoma. However, the rate of these concomitant mutations has been previously found least frequent as compared to *KRAS–TP53* mutations [45]. Moreover, we identified *EGFR* mutations on 6 different occasions with more common exon 20 insertions, while only on one occasion did we identify a variant in exon 21 or a deletion in exon 19. Exon 20 insertions are considered activating mutations that can cause approved TKIs to be ineffective [46,47]. Furthermore, *KRAS* non-synonymous mutations G15S and V7A appearing in concordance in our study were also detected to co-occur in gastric cancer patients who had been identified with therapeutic resistance, suggesting the importance of a new evaluation of the *KRAS* mutation and co-occurrence of other mutations’ effect in other cancers as well as lung cancer [48].

In line with previous studies [49,50], the changes in mutational burden were also observed during the disease progression in our study. Guo and colleagues reported decreasing VAF percentages (10% or more) for *EGFR* alterations in several patient samples after surgical treatment [51]. As reported here, the highest variation in VAFs was also observed in *EGFR* while, in the TX group, a 3-fold increased VAF for L858R was reported, indicating that not only mutation incidence but also allele frequency could be used in monitoring responses to treatment. Meanwhile, for most other gene mutations, the frequencies were below 5%, but even such low frequency variants could replicate the mutation profile of a tumor and, in a clinical setting, could influence therapy choices [52]. Mutation count increase from 2 to 4 in patients who had confirmed disease progression in the TX group could be an early indicative biomarker of metastasis occurrence or tumor heterogeneity. In line with VAF changes during disease progression, variation in mutation load could also be applied as a prognostic biomarker. Contrary to mutation frequencies, the mutation load was high in approximately half of the patients in T0 in our study, confirming that a high mutational burden is characteristic of the NSCLC and cfDNA load, and could be a valuable prognostic biomarker to predict disease progression [53]. As Hsu Y. C. and coauthors confirmed before, a high mutation load could be helpful in predicting relapse-free survival but not overall survival, with an estimated relapse-free survival of 4 years. However, in contrast to our study, this cohort in Hsu et al.’s study did not include *EGFR* mutant patients [54]. Moreover, similar to previous studies, we have also proven that increase in cfDNA load is a good predictor for disease progression [55,56]. In line with Gale and colleagues’ study, we detected elevated cfDNA load >5 months after the treatment as an indication of residual disease [57]. Furthermore, with increasing evidence of using the mutation load as a biomarker for immunotherapy resistant tumors, our study also confirms the importance of analyzing not only singular or concomitant mutation incidence but overall mutation load as a predictive biomarker for immunotherapy eligible late-stage NSCLC patient progression-free survival [58,59].

It should also be further recognized that this study has limitations as only 3 patients had samples collected at all three time points: before treatment (T0), after treatment (T1), and during disease progression (TX). Patient follow-up was increasingly more difficult as some patients did not return for specimen collection due to COVID-19 risk or due to the rapid disease progression. Despite this, the study allowed for a comprehensive comparison of cfDNA profiles in different disease phases. Next-generation sequencing was performed with the Ion Torrent™ Ion S5™ system using single-stranded DNA as a template, which heightens the rate of false positive calls. Moreover, cfDNA sequencing, according to the manufacturer, requires a high median read and molecular (>25,000 and 2500) coverage to identify rare mutations. Furthermore, validation of next-generation sequencing results was not possible due to the limited amount of sample volume and low concentrations of cfDNA.

## 4. Materials and Methods

### 4.1. Patients

From August 2018 to July 2020, 41 patients with NSCLC admitted to the National Cancer Institute in Vilnius, Lithuania, were enrolled in this prospective cohort study. All patients participating in this study were required to have provided appropriate written informed consent for the use of blood under regional bioethics committee regulations (permission No. 158200-18-993-496) (Table 1). Disease progression was defined when radiological progression was confirmed during follow-up. Overall survival was considered to be the time between the day of inclusion in the study to the last time that the patient’s medical records were reviewed. All death cases registered during the follow-up months to September 2021 (average 17.2, range 0.6–36.0 months) had a confirmed cause of death due to NSCLC.

### 4.2. Specimen Collection and Study Design

Blood samples (*n* = 72) for cfDNA mutation analysis from 41 patients were drawn from NSCLC patients at 3 predefined time points: (1) before treatment (*n* = 41; T0); (2) after treatment (15.4 ± 4.3 weeks after treatment initiation, *n* = 13; T1); (3) when disease progression was confirmed on subsequent reimaging (26.9 ± 3.6 weeks after treatment initiation, *n* = 18; TX). However, due to the COVID-19 pandemic and fast NSCLC progression, blood samples at all predefined time points were only successfully collected from 3 NSCLC patients. For 13 patients, no serial samples were available; for 13, serial samples were collected after the disease treatment; and for 15 patients, serial sample collection overlapped with the disease progression.

### 4.3. Mutational Analysis

Specimen preparation, cell-free DNA extraction, and next-generation sequencing library preparation, together with variant detection parameters, are detailed in Appendix B. Mutation detection was performed with Oncomine Lung cfDNA Assay using the Ion Torrent™ Ion S5™ system (Thermo Fisher Scientific (TFS), Waltham, MA, USA), while Ion Reporter Software v.5.12 (Ion Reporter Software, TFS) and the Integrative Genomics Viewer v.2.8.13 (Broad Institute, Cambridge, MA, USA) were used for mutation analysis.

### 4.4. Statistical Analysis

In our findings, the mutation count was used as a term to identify the number of mutations detected in any of our analyzed genes. In the survival prediction report, patients with 0, 1–2, and ≥3 mutations were categorized into groups according to mutation count. Mutation status was used as a metric to identify whether the mutation in the gene was detected or not. Variant allele frequencies were identified as low (<5%) and high (≥5%) in this study according to the rare somatic variant observed VAFs [60]. The median cfDNA load of 50 molecules per mL plasma in patients was used as a threshold to distinguish patients with high cfDNA load (>50 mutant cfDNA molecules) from those with no or low cfDNA load (≤50 mutant cfDNA molecules) according to Kruger and others [61]. Patient survival times were calculated from date of inclusion until progression or death.

Survival curves were generated using the Kaplan–Meier method and compared using the Log Rank Mantel–Cox tests. HRs and 95% CIs of the HRs were derived from the Cox proportional hazards models. A *p*-value below 0.05 was considered statistically significant. MedCalc version 14.8.1.0, R version 4.1.1 and GraphPad Prism version 8.01 were used for statistical analysis and reporting of the data collected for this study.

## 5. Conclusions

Hot spot mutations analysis in serial cfDNA samples from the Lithuanian NSCLC patient cohort confirmed the clinical value of *TP53* mutation and overall cfDNA mutational burden as a sensitive biomarker of the disease progression.

## Figures and Tables

**Figure 1 ijms-24-06958-f001:**
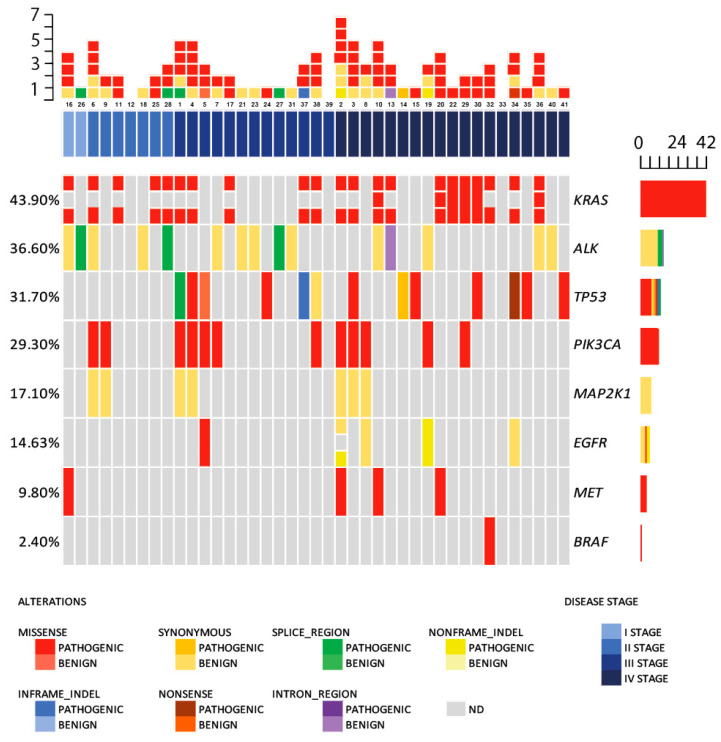
Summary of patients according to non-small cell lung cancer (NSCLC) stage and cell-free DNA (cfDNA) gene mutations in T0 group samples. Mutations are color-coded in gradient from pathogenic to benign after type of mutation: red (missense); blue (in-frame indel); orange (synonymous); brown (nonsense); green (splice site); violet (intronic region); yellow (non-frame indel). NSCLC patient stages are color-coded in blue gradient from darkest—IV stage to I stage. Grey colored squares represent non-detected (ND) mutations.

**Figure 2 ijms-24-06958-f002:**
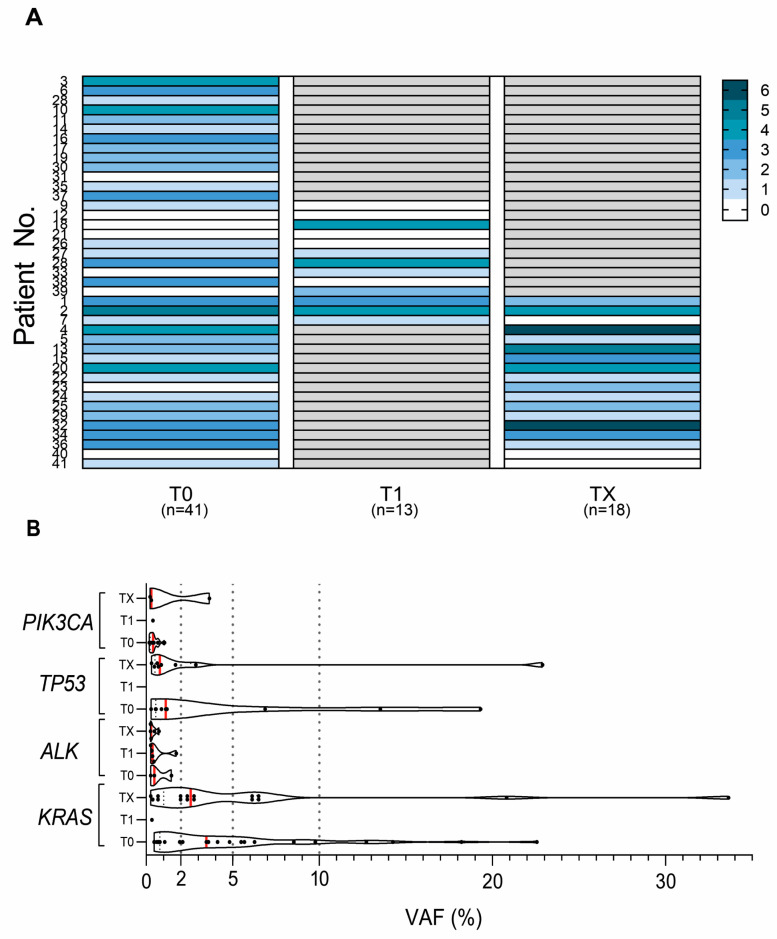
Changes in mutation count and variant allele frequency (VAF) according to sample collection time point in the T0, T1, and TX groups. (**A**) Mutation status and count distribution in T0, T1, and TX groups according to sample collection time; mutation count is color-coded in gradient from white—0 mutations to dark blue—6; grey—samples not collected. (**B**) Most frequently mutated actionable gene VAFs, according to sample collection time point in the T0, T1, and TX groups; each dot represents a sample, with violin plot corresponding to estimated density, and the median VAF is shown in a red vertical line; vertical dotted lines are at 2, 5, and 10% VAF.

**Figure 3 ijms-24-06958-f003:**
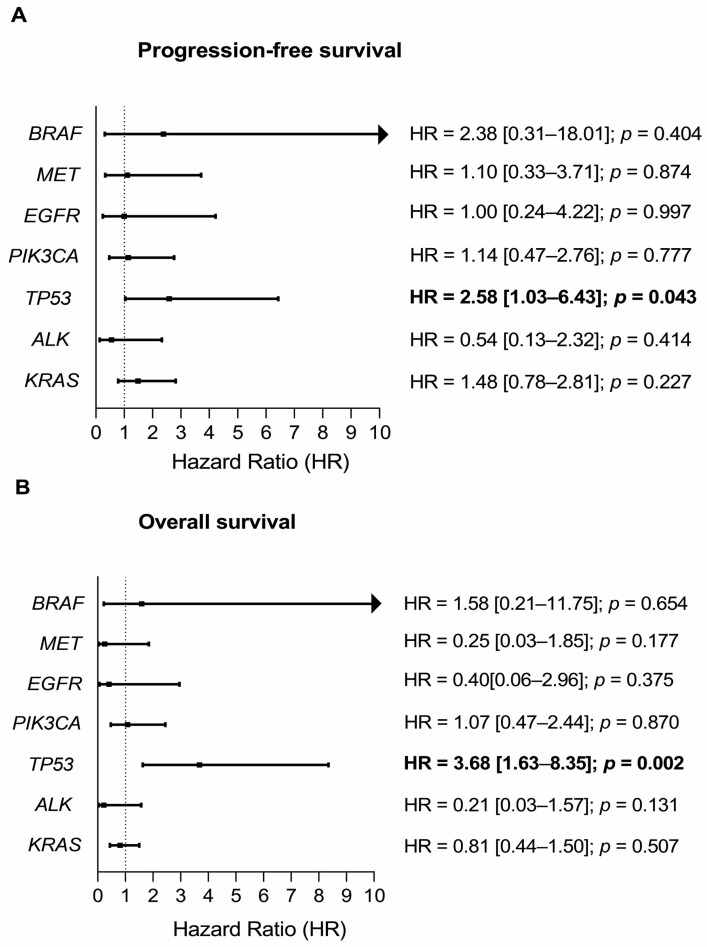
Hazard ratios (HR) [95% confidence interval (CI)] of genes with mutation status for progression-free survival (PFS) and overall survival (OS). (**A**) Progression free survival; (**B**) Overall survival.

**Figure 4 ijms-24-06958-f004:**
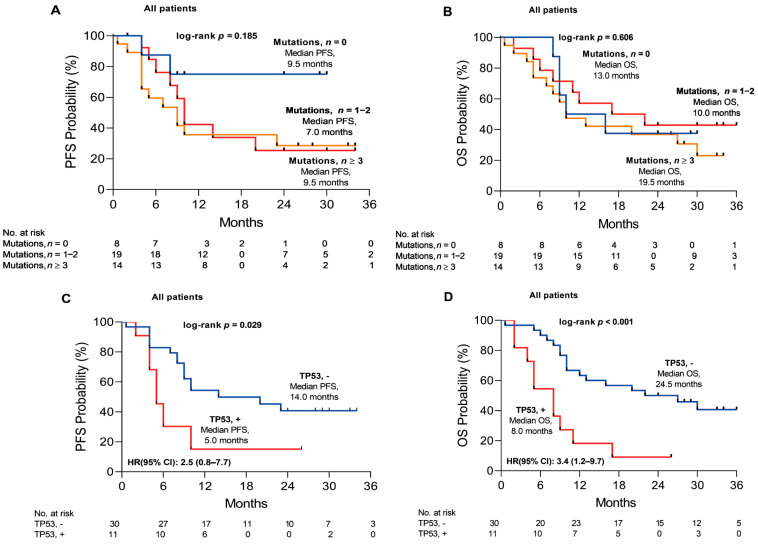
Kaplan–Meier estimates of progression–free and overall survival. (**A**,**B**) Patients categorized according to mutation count with 0, 1–2, and ≥3 mutations; mutation count is color-coded in blue—0, in orange—1−2 and in red—≥3 mutations. (**C**,**D**) Estimates according to mutant *TP53* status; blue color indicates negative *TP53* status and red—positive.

**Figure 5 ijms-24-06958-f005:**
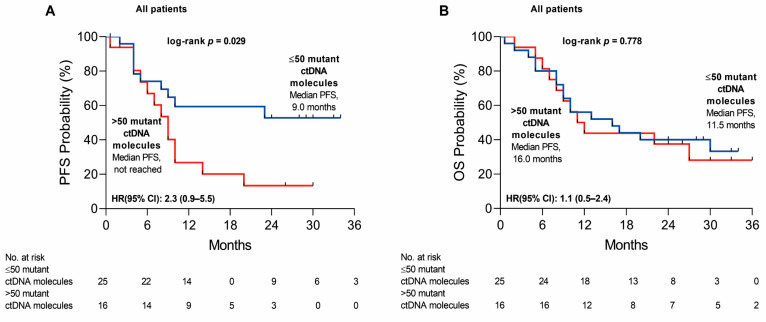
Kaplan–Meier estimates of progression-free survival (PFS) (**A**) and overall survival (OS) (**B**) according to median cfDNA load in high (>50 mutant cfDNA molecules) and low cfDNA load (≤50 mutant cfDNA molecules) patient groups; blue color indicates low cfDNA load (≤50 mutant cfDNA molecules) and red—high cfDNA load (>50 mutant cfDNA molecules).

**Table 1 ijms-24-06958-t001:** Clinical and demographic characteristics of patients with non-small cell lung cancer (NSCLC).

	NSCLC Patients (*n* = 41)
Age, median (range)	63 (44–78)
Sex, *n* (%)	
Male	31 (76)
Female	10 (24)
Histology, *n* (%)	
Adenocarcinoma	17 (41)
Squamous	17 (41)
Other	7 (17)
Disease stage, *n* (%)	
Ib	2 (5)
II	7 (17)
III	13 (32)
IV	19 (46)

## Data Availability

The data presented in this study are available on request from the corresponding author. The data are not publicly available (contain information that could compromise the privacy of the research participants).

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
