# Peer review of "Surveillance of cfDNA Hot Spot Mutations in NSCLC Patients during Disease Progression"

_ijms, 2023, doi:10.3390/ijms24086958_

Round 1
Reviewer 1 Report
The author include a high number of cases in early and advance stage . It could be necessary split the evolution and the alterations observed in.

Author Response
Thank you for your comments, we appreciate the time and effort that you have dedicated to providing your valuable feedback on our manuscript.
Response to Reviewer 1 Comments
Comment 1: The author includes a high number of cases in early and advance stage. It could be necessary split the evolution and the alterations observed in.
Response 1: Thank you for your comments, we appreciate the time and effort that you have dedicated to providing your valuable feedback on our manuscript. We agree that analyzing the mutations according to disease stage would be optimal, however our findings were not consistent with disease stage due to different types of mutations we detected. Few mutations that are targetable were more prominent in high variant allele frequency e.g. EGFR p.L858R (patient No. 5), MET p.T1010I (patient No. 16 and 18) in stage I and II of NSCLC patients and we did not detect correlations of variant allele frequency between non-small cell lung cancer stages (Supplementary Table 1). In this study we mainly focused on overall cfDNA as overall cfDNA load could be more generalized and more easily accessible way to predict progression-free and overall survival.
Reviewer 2 Report
This study proposes to analyze to identify and monitor lung cancer-specific mutations incidence in cell-free DNA as well as overall plasma cell-free DNA from 72 plasma samples of 41 patients. Four genes were detected with highest mutation incidence including KRAS, ALK, TP53, and PIK3CA. Among them, mutational status of TP53 and an overall cell-free DNA load were confirmed to be predictors of poor progression-free survival. The organization of this manuscript is very clear and finding are also interesting, however, the patient samples are not very complete with many missing tests, which may lead to some bias of the statistical results. I think some robust indicators can still be considered and useful in the future after filtering out some ambiguous and potentially bias results. Some minor opinions are listed below for authors’ consideration to improve the manuscript.
1. In introduction and discussion, there is not much comparison of the findings from past reports in the major mutations tested by cfDNA, more similar prior works need to be compared to specify the difference of the current findings/contributions to others.
2. In page 8, since this study has limitations as only 3 patients with samples collected at all three time points: before (T0), after treatment (T1) and during disease progression (TX), so will this give some bias on the statistical results about the rank or frequencies of the combined mutations to the disease progression prediction? If so, can we identify some more unaffected combined results which can give a robust guideline for better prognosis other than only the TP53 in the conclusion?
Author Response
Thank you for your comments, we appreciate the time and effort that you have dedicated to providing your valuable feedback on our manuscript. Please see the attachment.

Reviewer 3 Report
The objective of this study was to utilize plasma cfDNA analysis to identify and track the mutation profiles of the most prevalent lung cancer mutations in a Lithuanian NSCLC patient cohort. The study also aimed to evaluate the prognostic value of targeted next-generation sequencing for outcome prediction. The authors' research revealed that the KRAS, TP53, and PIK3CA gene mutations are the most common pathogenic variants found in NSCLC patients' cfDNA. This indicates that TP53 mutation and overall cfDNA mutational burden are useful biomarkers for monitoring disease progression and have clinical significance. The study is well-designed and executed, with clear descriptions of most of the methods. Some aspects require clarification, and some presented results lack high resolution. Nonetheless, despite these limitations, I recommend publication with minor revisions, as outlined in the authors' comments.
Major issues
1. Page 2 line 67: How authors select the 11 gene hotspot regions? Could you please clarify the criterion or references for selecting the gene hotspot regions.
2. Table 1: The study sample exhibits an imbalance in gender representation, with a disproportionate number of males and females. Additionally, the sample size for the Ib stage is limited, potentially leading to inadequate statistical power for data analysis. Therefore, achieving a more balanced gender distribution and increasing the sample size for the Ib stage could improve the study's statistical robustness and precision of the findings.
Minor issues
1. Figure 2A: Please provide a high-resolution figure. The original labels on the figure are blurred when zoomed in.
2. Page 7 line 202: “NCLC” typo.
Author Response

(The authors gave the same response as above.)

Round 2
Reviewer 1 Report
- Perhaps it would be advisable to include more patient cases and to have samples available at the three points of the follow-up.
- Did these patients have tissue identification of molecular biomarkers at diagnosis?
- It would be interesting to see if the driver mutations found at the diagnosis are maintained or if new resistance mutations have been found.
Author Response
We want to thank the Reviewer for their constructive comments and efforts towards improving our manuscript. The response to comment is uploaded as Word file. Please see the attachment.
